# Integration of Carbon Dioxide Removal (CDR) Technology and Artificial Intelligence (AI) in Energy System Optimization

Guanglei Li [1], Tengqi Luo [2], Ran Liu [3], Chenchen Song [4], Congyu Zhao [5], Shouyuan Wu [1,*] and Zhengguang Liu [6,*]

1 Key Laboratory of Power System Intelligent Dispatch and Control of Ministry of Education, Shandong University, Ji'nan 250061, China
2 College of Water Conservancy and Architectural Engineering, Northwest A&F University, Yangling 712100, China
3 Beijing National Laboratory for Condensed Matter Physics and Institute of Physics, Chinese Academy of Sciences, Beijing 100190, China
4 Higher Information Industry Technology Research Institute, Beijing Information Science and Technology University, Beijing 100192, China; scceducation@163.com
5 School of International Trade and Economics, University of International Business and Economics, Beijing 100029, China; cyzhao1998@163.com
6 Department of Chemical Engineering, The University of Manchester, Manchester M13 9PL, UK
* Correspondence: shouyuan_wu@sdu.edu.cn (S.W.); zhengguangliu@ieee.org (Z.L.)

**Abstract:** In response to the urgent need to address climate change and reduce carbon emissions, there has been a growing interest in innovative approaches that integrate AI and CDR technology. This article provides a comprehensive review of the current state of research in this field and aims to highlight its potential implications with a clear focus on the integration of AI and CDR. Specifically, this paper outlines four main approaches for integrating AI and CDR: accurate carbon emissions assessment, optimized energy system configuration, real-time monitoring and scheduling of CDR facilities, and mutual benefits with mechanisms. By leveraging AI, researchers can demonstrate the positive impact of AI and CDR integration on the environment, economy, and energy efficiency. This paper also offers insights into future research directions and areas of focus to improve efficiency, reduce environmental impact, and enhance economic viability in the integration of AI and CDR technology. It suggests improving modeling and optimization techniques, enhancing data collection and integration capabilities, enabling robust decision-making and risk assessment, fostering interdisciplinary collaboration for appropriate policy and governance frameworks, and identifying promising opportunities for energy system optimization. Additionally, this paper explores further advancements in this field and discusses how they can pave the way for practical applications of AI and CDR technology in real-world scenarios.

**Keywords:** climate change; low carbon; sustainable development; AI-CDR; 3E analysis



## 1. Introduction

### 1.1. Research Background

Carbon Dioxide Removal (CDR) technology and Artificial Intelligence (AI) are two prominent fields that have gained significant attention in recent years [1–3]. CDR technology focuses on the removal of carbon dioxide from the atmosphere [4], while AI encompasses the development of intelligent systems capable of performing tasks that typically require human intelligence [5,6]. The integration of CDR technology and AI presents a compelling opportunity to optimize energy systems and address the challenges of climate change. CDR technology aims to mitigate climate change by actively removing carbon dioxide from the atmosphere or capturing it from emission sources [7]. Various CDR methods have been proposed and developed including Direct Air Capture (DAC) [8], which extracts $CO_2$ directly from ambient air, and Bioenergy Carbon Capture and Storage (BECCS), which

combines biomass energy production with carbon capture and storage [9,10]. These technologies offer the potential to reduce carbon emissions and contribute to a more sustainable future. Figure 1 shows the important role of AI in CO$_2$ emission reduction. The left scheme corresponds to an AI-based GHG sequestrating cycle. The right scheme describes the role of AI software in this scheme.

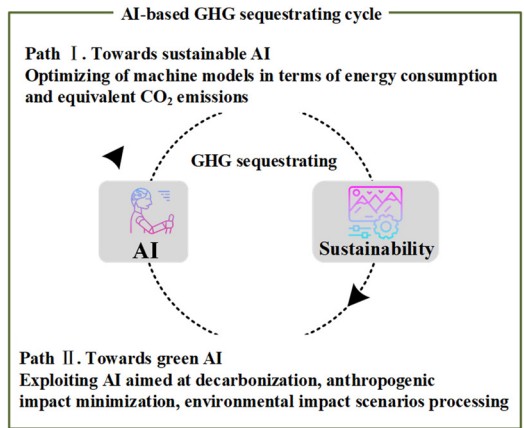 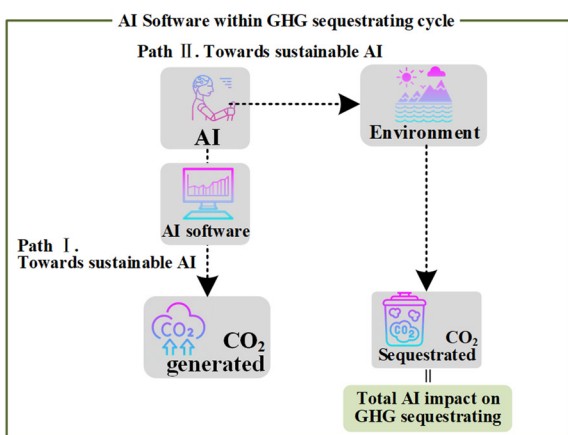

**Figure 1.** High-level schemes of AI-based GHG sequestrating.

Additionally, AI has demonstrated remarkable capabilities in data analysis, pattern recognition, and optimization [11]. By harnessing the power of AI, energy systems can be optimized to achieve greater efficiency, reduce the carbon footprint, and improve overall performance. The integration of CDR technology and AI holds significant promise for energy system optimization. AI techniques can enhance CDR technologies in several ways including accurate assessment and monitoring of carbon emissions, optimization of deployment and operation of CDR facilities, and real-time monitoring and adaptive control based on changing conditions [12].

The motivation behind integrating CDR technology and AI lies in the urgent need to address climate change and transition to a sustainable energy future. The rising levels of carbon dioxide in the atmosphere and the associated environmental consequences necessitate effective strategies to reduce emissions and remove existing carbon [13]. By combining CDR technology and AI, we can enhance the efficiency and effectiveness of carbon removal processes, optimize energy systems, and accelerate the transition to a low-carbon economy. The integration of CDR technology and AI presents a unique opportunity to optimize energy systems and mitigate climate change. By combining the capabilities of CDR technology in carbon removal and AI in data analysis and optimization, we can achieve more efficient and sustainable energy systems. The following sections of this review will delve deeper into the applications, advantages, challenges, and future prospects of this integration.

### 1.2. Overview of CDR Technology

Carbon Dioxide Removal (CDR) technology encompasses various methods and approaches aimed at removing carbon dioxide from the atmosphere or capturing it from emission sources. Here are some common CDR technologies.

- Direct Air Capture (DAC)

Principle: DAC technology directly extracts carbon dioxide from ambient air using chemical sorbents or membranes [14]. Application Scope: DAC can be deployed in various locations including industrial sites, power plants, or even directly in the atmosphere. Potential Challenges: DAC technologies face challenges related to energy consumption, cost-effectiveness, and scalability. Developing efficient and cost-competitive sorbents and optimizing the capture process are ongoing areas of research. Figure 2 shows the levelized cost projections for CO$_2$ direct air capture in 2050.

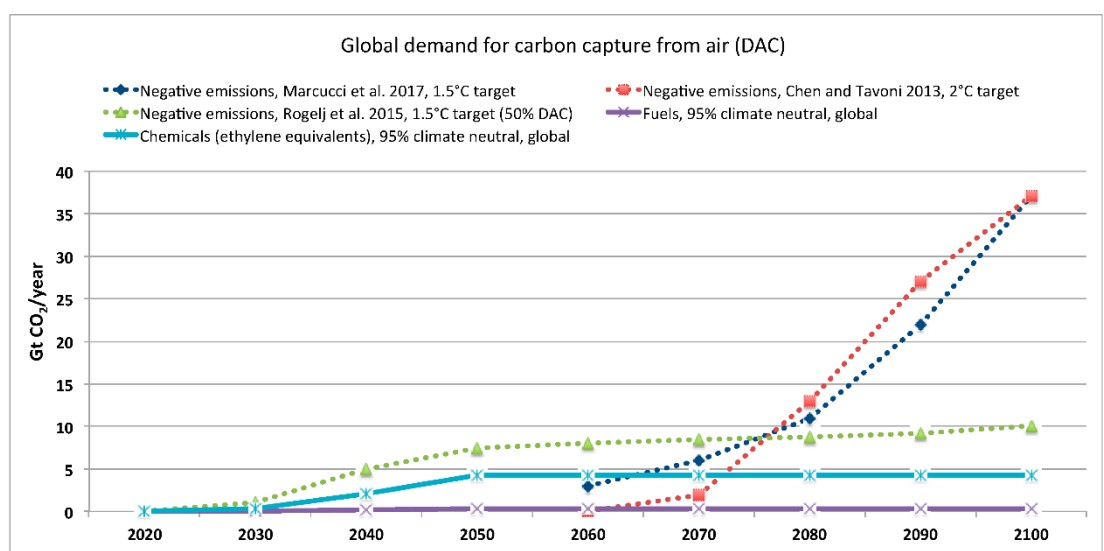

**Figure 2.** Global Direct Air Capture (DAC) demand for achieving negative emissions from 2060 in comparison with globally required climate-neutral $CO_2$ for Power-to-Liquids and Power-to-Chemicals in Gt $CO_2$/year [15–18].

- Bioenergy Carbon Capture and Storage (BECCS)

Principle: BECCS combines biomass energy production with carbon capture and storage. The process flow diagram in Figure 3 depicts the operations during the harvest season, which involves steam extraction for industrial processes and includes CCS. Biomass plants capture $CO_2$ emissions generated during energy production and store them underground [19]. Application Scope: BECCS can be implemented in power plants, industrial facilities, or dedicated biomass production facilities. Potential Challenges: Challenges include ensuring sustainable biomass feedstock supply, optimizing the energy balance of the process, and addressing potential environmental impacts associated with large-scale biomass cultivation.

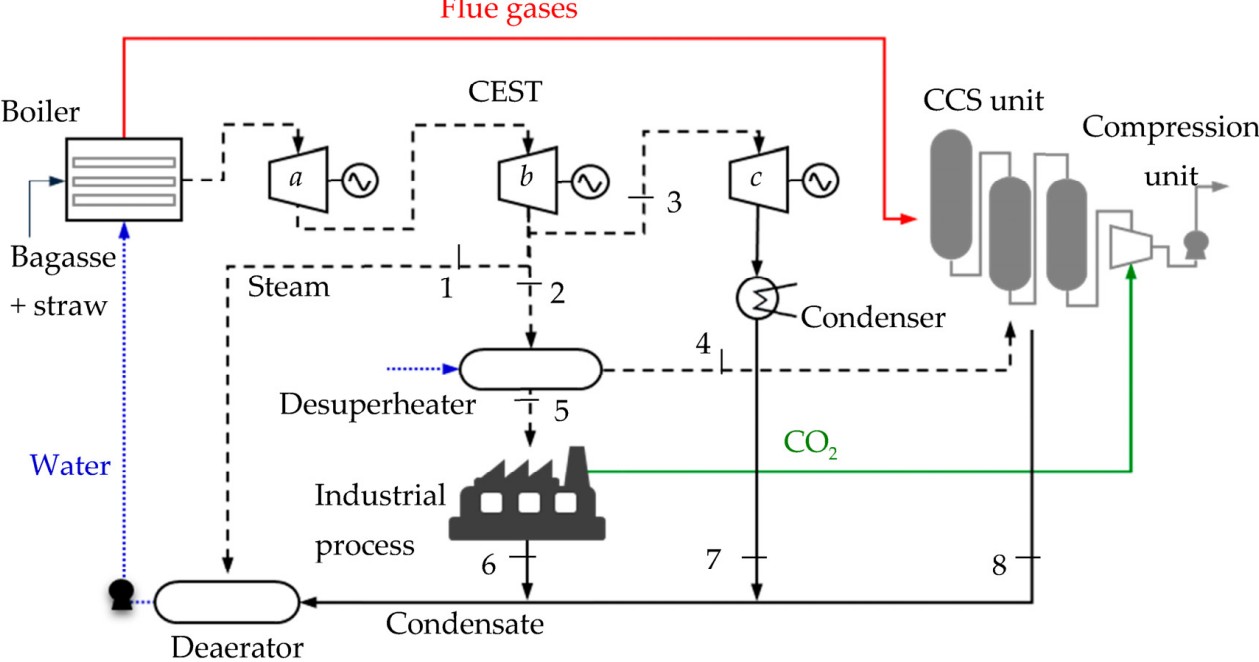

**Figure 3.** BECCS process flow diagram (harvest season) [20].

- Carbon Sequestration

  Principle: Carbon sequestration involves capturing and storing carbon dioxide emissions in geological formations, such as depleted oil and gas reservoirs or deep saline aquifers [21]. Application Scope: Carbon sequestration can be applied to various emission sources including power plants and industrial facilities. Potential Challenges: Challenges include selecting suitable storage sites, ensuring the long-term integrity of the storage reservoirs, and addressing public perception and regulatory concerns surrounding the safety and permanence of storage.

- Enhanced Weathering

  Principle: Enhanced weathering involves accelerating natural weathering processes to capture and store carbon dioxide [22]. It typically involves the application of minerals or rocks that react with $CO_2$ and permanently sequester it. Figure 4 illustrates a conceptual diagram that presents the strategies of CCS through mineral carbonation. Application Scope: Enhanced weathering can be applied to agricultural lands, coastal areas, or specific carbon capture facilities. Potential Challenges: Challenges include identifying suitable mineral sources, assessing the environmental impact of large-scale mineral deployment, and understanding the long-term stability of carbon storage.

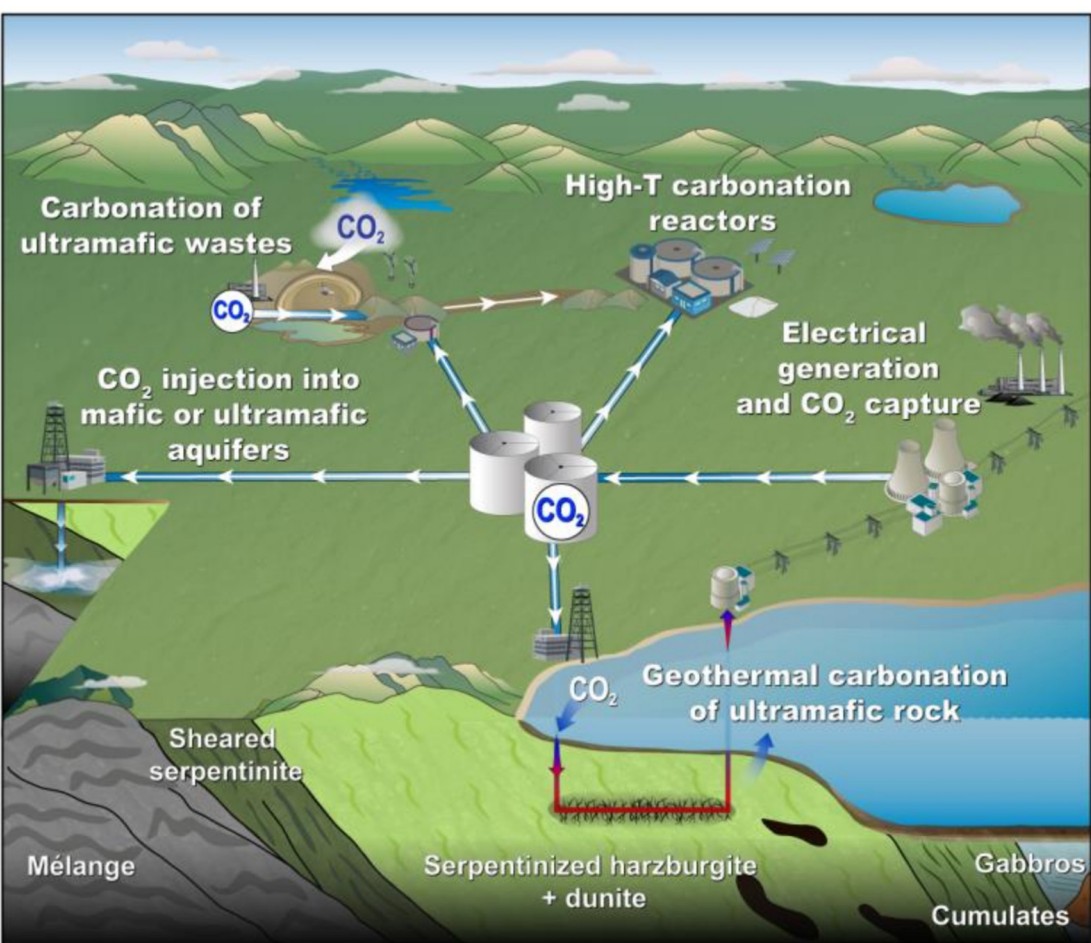

**Figure 4.** Visual graphics of enhanced weathering processes [23].

Each CDR technology has its own set of principles, application scopes, and challenges. It is crucial to evaluate these factors to determine the most suitable CDR technology for a given context. The advantages and disadvantages of each technology are compared as shown in Table 1.

Table 1. Comparison of advantages and disadvantages of the four technologies.

| Technologies | Advantages | Disadvantages |
|---|---|---|
| DAC | 1. It can capture carbon dioxide from the air at any location. 2. Potential for large-scale carbon capture and storage. | 1. It requires significant amounts of electricity and energy. 2. Relatively high cost compared to traditional methods. |
| BECCS | 1. It utilizes biomass feedstocks, which can be renewable and sustainable. 2. It can be integrated with existing power plants or industrial facilities. | 1. It requires large areas of land to cultivate biomass feedstocks. 2. It competes with land use for food production and conservation efforts. |
| Carbon Sequestration | 1. It enhances soil fertility and promotes sustainable land management practices 2. It can contribute to the development of carbon offset projects | 1. Limited capacity to sequester large amounts of carbon dioxide 2. High costs associated with implementation and maintenance |
| Enhanced Weathering | 1. It has the potential to improve soil fertility and agricultural productivity 2. It does not require complex infrastructure for implementation | 1. High costs associated with large-scale implementation and transportation of materials 2. Potential challenges related to land use and ecosystem disruption |

### 1.3. Research Gap of Past Reviews

While there have been previous literature reviews on Carbon Dioxide Removal (CDR) technology and Artificial Intelligence (AI) individually, there is a research gap when it comes to exploring their integration specifically for energy system optimization [24]. Previous reviews have predominantly emphasized either CDR technologies or AI applications in energy systems with limited consideration of their combined potential. This review aims to address this gap by examining the integration of CDR technology and AI, highlighting their synergistic effects and advantages for energy system optimization.

By integrating CDR technology and AI, we can optimize energy systems and tackle climate change challenges more effectively. While there have been studies on optimizing energy systems using AI techniques, there is a lack of research specifically focused on integrating CDR technology into these optimization strategies. This review aims to fill this gap by exploring how AI can optimize the deployment, operation, and performance of CDR technologies within energy systems.

Real-time monitoring and adaptive control are essential when integrating CDR technology and AI. AI enables the real-time monitoring of CDR facilities, allowing for adaptive control and optimization based on changing conditions and demands. This capability enhances the efficiency and effectiveness of CDR technology in addressing carbon emissions. The integration of CDR technology and AI has significant implications for sustainable energy development. It contributes to carbon emission reduction, improved energy system efficiency, and the development of new technologies, supporting the transition to a sustainable energy future.

In summary, this review provides a comprehensive analysis of the integration of CDR technology and AI. It covers the fundamental concepts and principles of CDR technology and AI, explores integration methods and potential advantages, and discusses practical challenges, feasibility, and environmental management aspects. By focusing on a specific gap in research and highlighting the advantages of this review, we aim to provide valuable insights for researchers, policymakers, and practitioners in the field of energy system optimization and sustainable development.

### 2. Integration of "CDR + AI"

The integration of CDR Technology and Artificial Intelligence (AI) offers several potential advantages, including four parts as follows:

### 2.1. Accurate Carbon Emissions Assessment

By combining CDR technology and AI, it becomes possible to accurately assess and quantify carbon emissions. Figure 5 shows an overview of the thematic dimensions included in the technical feasibility assessment of CDR removal. AI algorithms can analyze large amounts of data from various sources, including CDR facilities, industrial processes, and energy consumption patterns, to provide more precise estimations of carbon emissions. This can help in identifying high-emission areas, tracking progress towards emission reduction goals, and informing policy decisions [25].

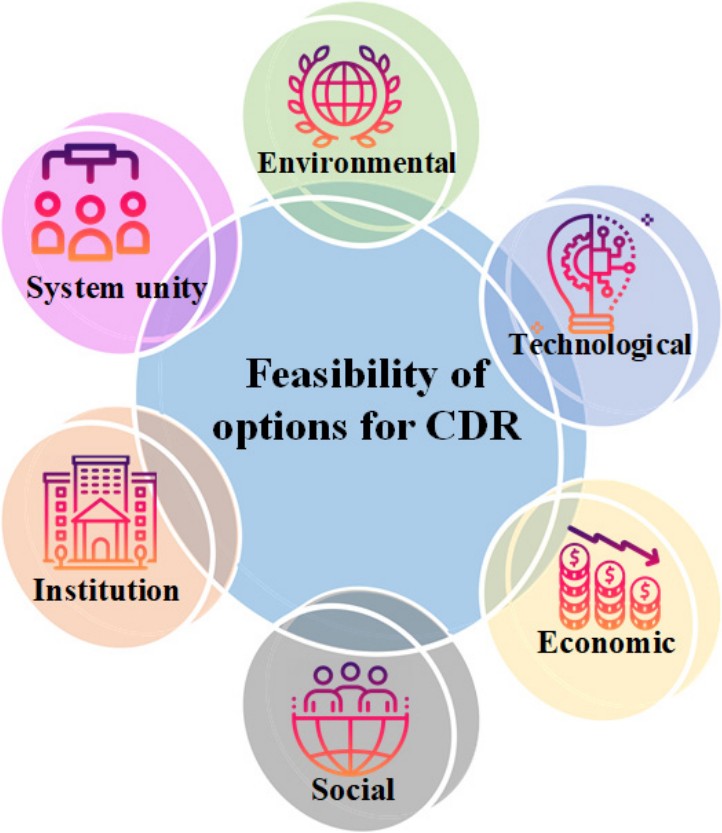

**Figure 5.** Overview of thematic dimensions included in the feasibility assessment framework of CDR options.

AI algorithms can integrate and analyze data from multiple sources, including emissions inventories, satellite imagery, sensor networks, and industry-specific data, to gain a comprehensive understanding of emissions across different sectors and regions [26]. By analyzing emissions-related data, AI can identify the spatial and temporal patterns of emissions, such as emission hotspots or areas with significant emission fluctuations, and pinpoint the high-emission areas that may require targeted interventions [27]. Machine learning techniques enable AI models to recognize emission patterns and make predictions based on historical data [28], identifying areas with a higher likelihood of being high-emission areas [29,30].

Satellite imagery provides valuable information on greenhouse gas concentrations, land-use changes, and industrial activities [31,32], which AI algorithms can process and analyze to identify regions with higher emissions and track changes over time [33]. Integrating data from sensors and IoT devices enables real-time monitoring of emissions, facilitating the identification of areas experiencing sudden spikes or persistent high emissions [34,35]. By visualizing emissions data spatially and using geospatial analysis techniques, AI can provide intuitive representations of high-emission areas [36,37]—making it easier for policy-makers and stakeholders to identify regions that require targeted mitigation strategies [38].

By leveraging these capabilities, AI algorithms can assist in identifying high-emission areas, providing valuable insights into the sources and patterns of emissions. This information can guide policymakers in developing targeted interventions, implementing emission reduction measures, and prioritizing areas for mitigation efforts.

### 2.2. Optimized Energy System Configuration

AI algorithms can optimize the integration of CDR technology into energy systems by analyzing data on energy demand, renewable energy generation [39], and other factors to identify the most efficient and cost-effective ways to reduce carbon emissions and improve overall system efficiency [40,41]. By maximizing the use of renewable energy sources, AI can help reduce carbon emissions. Figure 6 is a schematic of a low-carbon energy system.

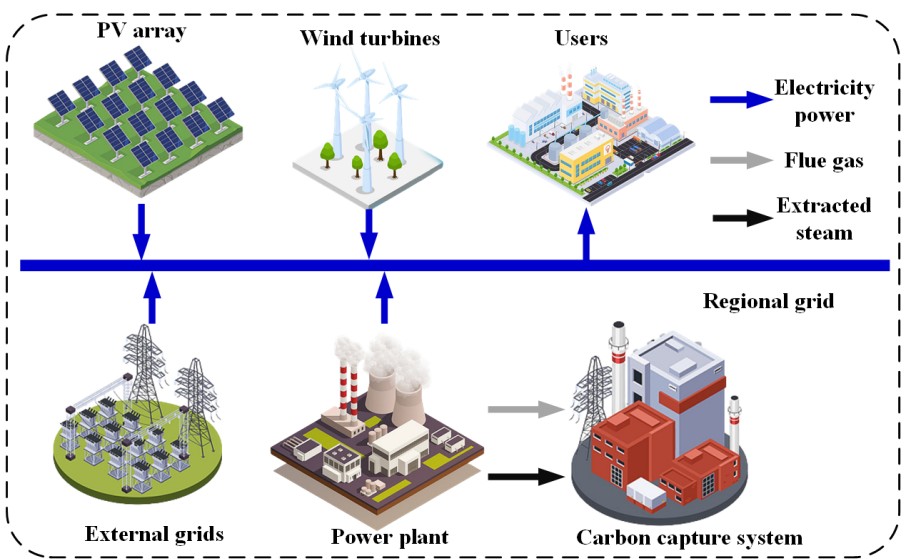

**Figure 6.** The schematic diagram of a low-carbon energy system.

AI algorithms can analyze extensive datasets related to energy demand, renewable energy generation, grid infrastructure, and other relevant factors [42,43]. By processing this data, AI models can build sophisticated models that capture the complexities of an energy system including the interplay between different energy sources, demand patterns, and carbon emissions [40,44]. These models enable scenario analysis and optimization to identify efficient and cost-effective configurations for integrating CDR technology [45]. Considering factors such as energy demand, renewable energy availability, storage capacities, and carbon removal targets, AI algorithms simulate and evaluate different system configurations [46,47]. This facilitates the identification of optimal solutions that maximize renewable energy use, minimize carbon emissions, and achieve specific energy and carbon removal objectives.

AI algorithms also optimize demand-side management strategies by analyzing energy demand patterns [12,48]. By leveraging machine learning techniques, AI identifies demand response opportunities, predicts peak energy demand periods, and optimizes the scheduling of energy-consuming activities [49,50]. This helps balance energy supply and demand, reduce reliance on fossil fuel-based energy generation, and increase the integration of renewable energy and CDR technologies. AI enhances the accuracy of renewable energy forecasting by analyzing historical weather data, renewable energy generation data, and other variables [51,52]. Accurate predictions of renewable energy availability enable the optimization of CDR facility scheduling and operation, aligning them with high renewable energy generation and low grid demand [53,54]. Furthermore, AI algorithms optimize energy system configurations by analyzing historical and real-time data on energy supply and demand, market prices, weather conditions, and other factors [55]. This analysis identifies opportunities for energy storage deployment, demand shifting, and smart grid manage-

ment [56], ensuring stability, accommodating intermittent renewable energy sources, and effectively integrating CDR technologies [39].

By leveraging AI capabilities, energy system operators, policymakers, and stakeholders can optimize energy system configurations to maximize CDR technology benefits. This includes minimizing carbon emissions, maximizing renewable energy use, and improving overall system efficiency and resilience.

### 2.3. Real-Time Monitoring and Scheduling of CDR Facilities

AI enables the real-time monitoring and adaptive control of CDR facilities. By analyzing data from sensors, AI algorithms can continuously monitor the performance and operation of CDR facilities, detecting any anomalies or inefficiencies [57]. This allows for timely adjustments and optimizations, ensuring optimal utilization of resources and maximization of the carbon removal capacity of the facilities. Figure 7 displays a map depicting the global distribution of CCUS facilities with a specific focus on Europe.

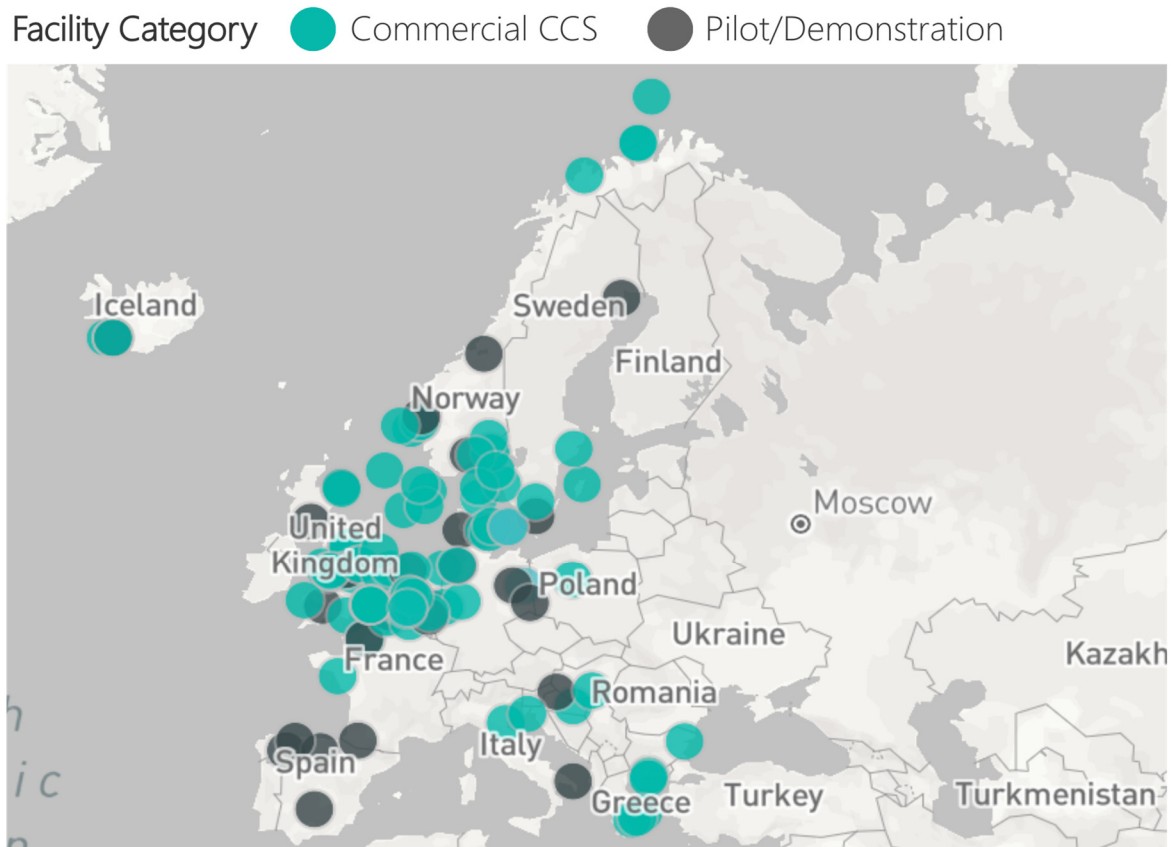

**Figure 7.** Worldwide distribution of CCUS facilities divided by categories, expanded in Europe [58].

AI algorithms integrate data from various sensors and monitoring devices installed in CDR facilities [59,60]. This includes parameters like temperature, pressure, flow rates, and capture efficiency. By continuously analyzing real-time data, AI monitors the performance of CDR facilities, detects anomalies or deviations from optimal conditions, and alerts operators to potential issues [61,62]. Anomaly detection techniques help identify abnormal behavior or malfunctions [63], triggering alarms or notifications by comparing real-time sensor data with historical patterns and predefined thresholds. Operators can take immediate corrective actions, minimizing disruptions in the carbon removal process. AI also predicts maintenance needs and schedules proactive maintenance activities, optimizing facility availability and reliability [64].

AI dynamically adjusts CDR facility operations based on real-time data and changing conditions. By monitoring factors like energy availability, carbon capture efficiency, and

storage capacity, AI optimizes scheduling and resource allocation. This enables adaptive control strategies that maximize carbon removal capacity, optimize energy consumption, and respond to fluctuations in renewable energy generation or demand. AI optimizes resource allocation within CDR facilities, considering real-time data on energy availability, cost, and carbon removal targets. This determines the most efficient allocation of resources for optimal carbon removal performance, minimizing costs while maximizing capacity. AI integrates with energy grid data and market signals to schedule CDR facilities. Considering electricity prices, demand peaks, and renewable energy availability, AI schedules carbon removal processes during periods of low electricity demand or high renewable energy availability. This maximizes renewable energy utilization, reduces costs, and aligns carbon removal activities with grid conditions.

### 2.4. Mutual Benefits and Mechanisms

The integration of CDR technology and AI can lead to mutual benefits and synergies [65]. CDR technology can provide data for AI model training and improvement, while AI optimization methods can improve the efficiency of CDR technology. The data collected from CDR facilities can be used to train AI models and enhance their accuracy and efficiency. AI algorithms can analyze complex datasets and optimize the operation and performance of CDR facilities, leading to increased carbon removal efficiency and reduced operational costs [22].

CDR facilities generate a wealth of data that can be utilized to train AI models [3]. By incorporating this data into the training process, AI algorithms can learn from real-world CDR operations and improve their accuracy and efficiency. This leads to more effective AI models that can make better decisions and optimizations in CDR technology [66].

AI algorithms can optimize the operation and performance of CDR facilities by analyzing complex datasets and identifying patterns and correlations [67]. This enables AI to make informed decisions and adjustments in real-time, enhancing the efficiency and effectiveness of CDR technology. AI algorithms can optimize various aspects of CDR technology processes, including capture, storage, and utilization of carbon dioxide, leading to cost reductions, energy savings, and increased carbon removal capacity.

AI algorithms can enable CDR systems to be adaptive and responsive to changing conditions by continuously analyzing real-time data. This adaptability allows CDR systems to optimize their performance in response to variations in energy supply, carbon emissions, and other relevant factors, ensuring effective carbon removal in real-time [68]. AI can also play a crucial role in planning the deployment and scalability of CDR technology by analyzing various factors and optimizing the allocation of resources [69].

The integration of CDR technology and AI creates a symbiotic relationship that enables improved carbon removal capabilities, cost-effectiveness, and scalability. This synergy contributes to the mitigation of climate change by offering accurate carbon emissions assessments, optimized energy system configurations, and real-time monitoring and scheduling of CDR facilities. The mutual benefits between CDR technology and AI can drive advancements in both fields, leading to more efficient and effective carbon removal solutions.

## 3. E Performance Improvement for Energy System

The integration of CDR technology and AI can indeed result in performance improvements in terms of efficiency, environmental impact, and economic viability [70]. The integration of CDR technology and AI can bring about significant performance improvements in terms of efficiency, environmental impact reduction, and economic viability [71]. These improvements contribute to the advancement and adoption of CDR technology as a crucial tool in addressing climate change and achieving sustainable carbon mitigation goals.

### 3.1. Efficiency Improvements

AI algorithms can optimize the operation and processes of CDR facilities, leading to increased efficiency. By analyzing large volumes of data, AI models can identify patterns,

correlations, and optimal operating conditions. This optimization can improve the overall efficiency of carbon removal processes, such as capture, storage, and utilization, reducing energy consumption and resource wastage [72]. Enhanced efficiency translates to higher carbon removal rates per unit of energy or resources utilized. Figure 8 shows how compression energy consumption varied with cooling temperature and compressor discharge pressure for three $CO_2$ capture scenarios.

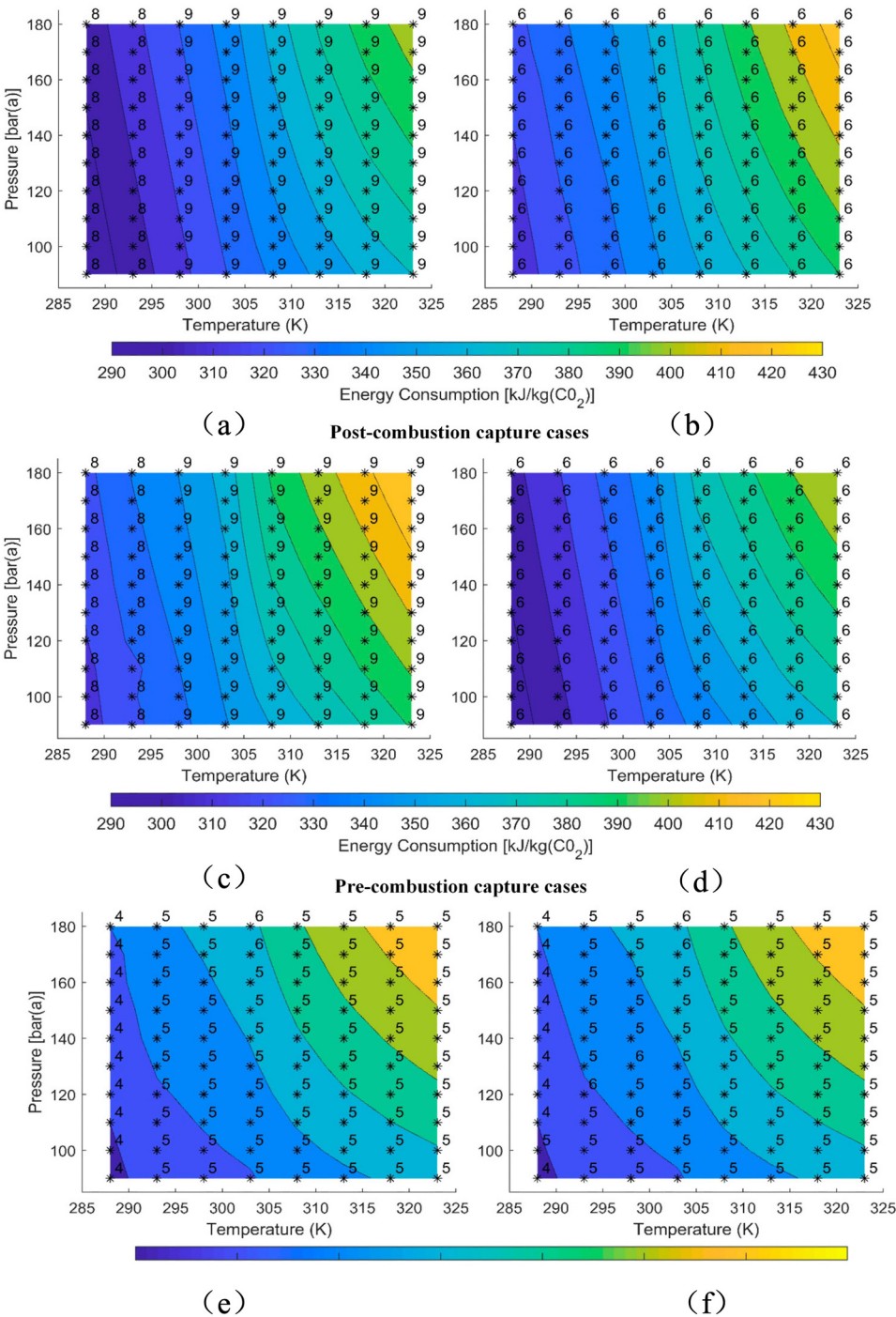

**Figure 8.** Three cases of optimized compression energy consumption and number of stages. (**a**,**b**) Post-combustion capture cases; (**c**,**d**) Pre-combustion capture cases; (**e**,**f**) Oxyfuel cases [73].

AI algorithms analyze data generated by CDR systems including sensor readings, operational parameters, and historical performance data. By processing this data, AI models identify patterns, correlations, and optimal operating conditions that maximize carbon

removal efficiency [74]. For example, AI determines the optimal flow rates, temperature, pressure, and solvent concentrations for carbon capture processes, minimizing energy consumption and improving capture efficiency. AI enables real-time monitoring and adaptive control of CDR facilities, detecting deviations from optimal conditions and making necessary adjustments [75].

AI optimizes energy consumption within CDR facilities by analyzing energy usage patterns and considering factors like costs and availability. This includes strategies such as load balancing, scheduling energy-intensive operations during low-demand periods, and integrating renewable energy sources [76]. AI also optimizes resource allocation and utilization within CDR facilities by analyzing data on availability, costs, and carbon removal targets. This minimizes waste, reduces costs, and maximizes the facility's carbon removal capacity. AI aids in the early detection of equipment faults or maintenance needs within CDR systems [77]. By continuously monitoring sensor data and performance metrics, AI identifies anomalies or deviations that may indicate impending failures. This enables proactive maintenance planning and reduces downtime, ensuring efficient and continuous facility operation. AI continuously learns and adapts based on feedback and performance data, identifying areas for improvement and refining optimization strategies over time [78]. This ongoing learning process leads to efficiency improvements and the better performance of CDR systems.

By leveraging AI algorithms to optimize operations, control parameters, and allocate resources, CDR facilities achieve higher efficiency, reducing energy consumption and resource wastage while maximizing carbon removal rates. These improvements contribute to the effectiveness of CDR technology in mitigating carbon emissions and combating climate change.

### 3.2. Environmental Impact Reduction

AI optimization methods can significantly reduce the environmental impact of CDR technology by improving the efficiency of carbon removal processes and minimizing energy requirements and associated greenhouse gas emissions [79,80]. Additionally, AI algorithms can identify opportunities for utilizing captured carbon dioxide in industrial processes or carbon utilization technologies, further reducing emissions and environmental impact [81].

By optimizing the energy consumption of CDR processes, AI algorithms can minimize energy usage without compromising carbon removal efficiency, thus reducing the carbon footprint of CDR facilities. AI can also optimize various carbon removal processes, such as capture, storage, and utilization, leading to more efficient and environmentally friendly operations [3,82]. For instance, AI algorithms can determine optimal operating conditions for carbon capture technologies, such as solvent selection, temperature, and pressure, to maximize efficiency and minimize energy consumption, thereby reducing the environmental impact of carbon removal.

Furthermore, AI algorithms can help identify opportunities for utilizing captured carbon dioxide, such as in enhanced oil recovery, carbon-based materials production, or carbon mineralization, thereby further reducing the environmental impact of CDR [83]. AI can also facilitate comprehensive life cycle assessments to evaluate the environmental impact of CDR processes comprehensively, ensuring that environmental considerations are taken into account throughout the CDR system's life cycle. Additionally, AI can assist in environmental monitoring and compliance in CDR facilities by detecting deviations from regulatory standards or environmental thresholds, enabling timely corrective actions to mitigate potential environmental impacts and ensure compliance with environmental regulations [63,84].

In summary, the integration of CDR technology with AI can lead to a substantial reduction in environmental impact through energy optimization, process efficiency improvements, identification of carbon utilization opportunities, comprehensive life cycle assessments, environmental monitoring, and climate change mitigation. These environ-

mental benefits highlight the potential of CDR technology integrated with AI in addressing climate change and promoting a sustainable future.

*3.3. Economic Feasibility*

The integration of AI can enhance the economic viability of CDR technology by identifying cost-effective strategies and optimizing resource allocation, energy management, and process efficiency [85]. This optimization leads to reduced operational costs, making CDR technology more economically feasible and attractive for implementation, fostering its adoption and scalability [86].

AI algorithms can optimize the allocation of resources and energy consumption within CDR facilities, leading to cost savings [87]. They can also assist in the scalability and deployment planning of CDR technology and provide decision support for stakeholders considering investments in CDR technology [88]. By optimizing costs, resource allocation, energy management, and process efficiency, AI integration enhances the economic viability of CDR technology [89], making it more attractive for implementation, fostering its adoption and scalability, and facilitating its role in addressing climate change and achieving sustainable carbon mitigation goals. Figure 9 shows the relationship between the net present value of negative $CO_2$ and the cost of biomass (fuel) in both scenarios.

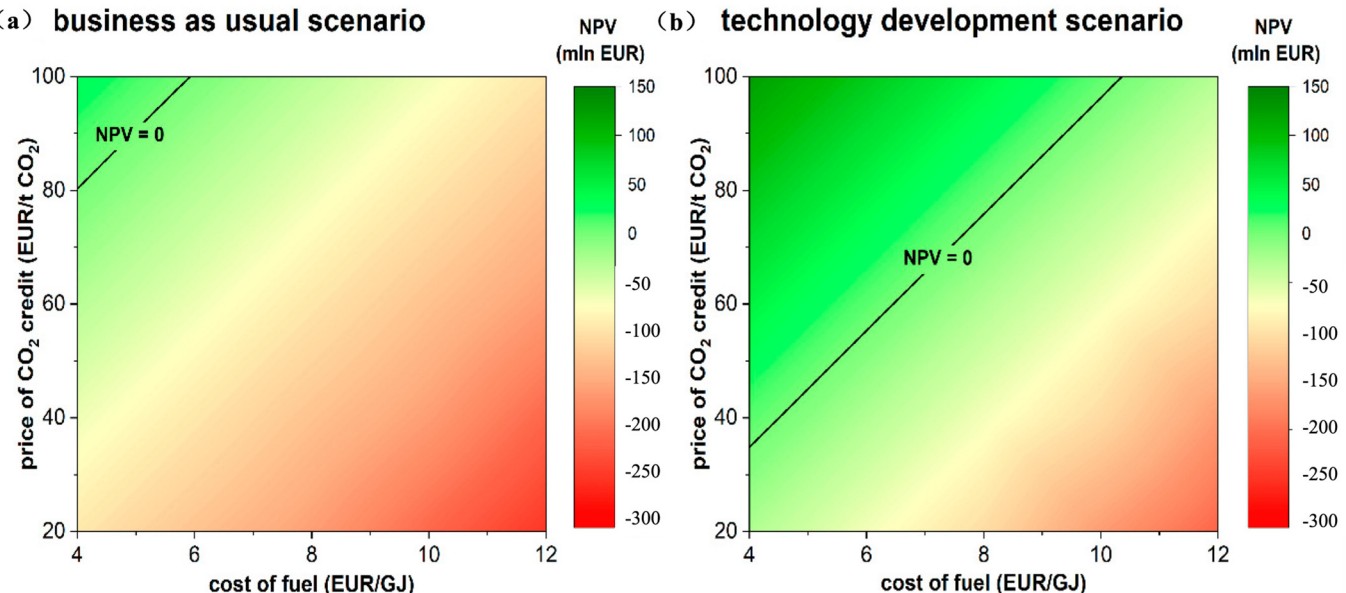

**Figure 9.** NPV as a function of negative $CO_2$ emission credits and the cost of fuel in (**a**) the business-as-usual scenario and (**b**) the technological development scenario [90].

## 4. Perspectives and Challenges

The success of AI algorithms depends on the availability and quality of data. Ensuring data accuracy, consistency, and reliability is crucial for accurate modeling and optimization. AI models used in CDR optimization may have inherent uncertainties due to the complex and dynamic nature of environmental conditions. Addressing data quality issues and accounting for model uncertainty is essential for reliable decision-making in CDR technology, as shown in Figure 10. Evaluating the scalability and applicability of AI algorithms in different CDR processes and system configurations is necessary to ensure technological feasibility. Legal and regulatory challenges, as well as public perception and acceptance, also need to be addressed for successful integration and adoption.

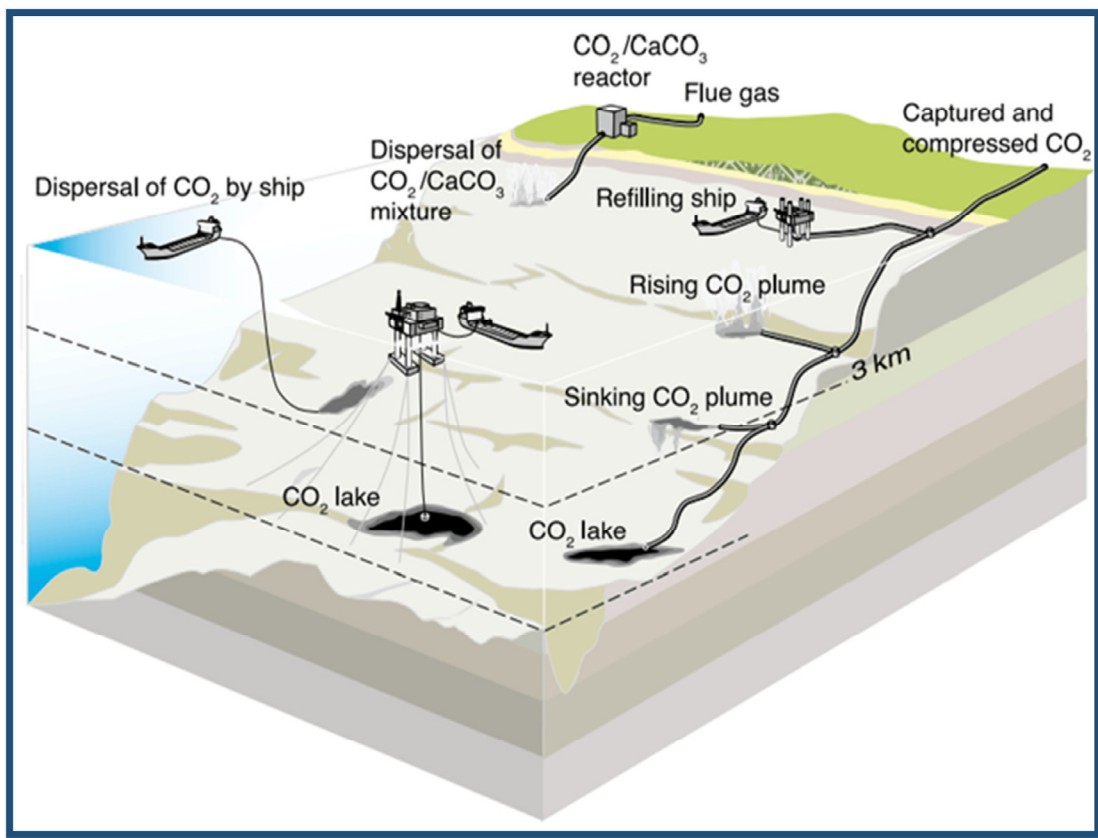

**Figure 10.** Schematic of several ocean storage strategies. IPCC Special Report on Carbon Dioxide Capture and Storage [91].

Future Research Directions and Key Focus Areas: Future research should focus on developing advanced AI models and optimization algorithms specifically tailored for CDR technology, including addressing model uncertainties and improving data collection methods. Developing decision support tools that account for uncertainties and risks associated with CDR technology and AI integration is essential. Collaboration between researchers, engineers, policymakers, and stakeholders is crucial for addressing technical, economic, regulatory, and social challenges. Research should also explore the development of policy and governance frameworks that support the integration of CDR technology and AI and conduct large-scale demonstrations and field trials of integrated CDR technology and AI systems.

By addressing challenges related to data quality, model uncertainty, technological feasibility, policy and regulatory environments, and public acceptance and focusing on key research areas, the integration of CDR technology and AI can advance energy system optimization for carbon removal, contributing to the practical implementation and widespread adoption of CDR technology to mitigate climate change and achieve sustainable carbon mitigation goals.

## 5. Conclusions

Through this review, we have gained a comprehensive understanding of the applications and potentials of integrating CDR technology and AI in energy system optimization. This integration offers new solutions for sustainable energy development, contributing to carbon emission reduction, improved energy system efficiency, and the development of new technologies.

1. This paper provides a comprehensive summary of the integration of AI and CDR technology in energy system optimization. AI optimization algorithms identify cost-effective strategies, such as optimal resource allocation, energy management, and process

optimization, reducing operational costs and promoting the adoption and scalability of CDR technology.

2. This review outlines four approaches to integrating CDR technology and AI including using AI optimization algorithms for resource allocation, energy management, process efficiency, and decision support. These approaches enhance the efficiency and effectiveness of CDR technology in carbon reduction, contributing to sustainable energy system optimization.

3. The integration of AI and CDR technology positively impacts three aspects of energy systems: environment, economy, and energy. AI optimization algorithms reduce energy consumption and carbon emissions, mitigating environmental effects. By optimizing costs, CDR technology becomes more economically feasible, reducing operational expenses. AI optimization algorithms also improve energy utilization efficiency, promoting sustainable energy development.

4. This paper suggests future research directions and areas of focus, such as improving modeling and optimization techniques, enhancing data collection and integration capabilities, enabling robust decision-making and risk assessment, fostering interdisciplinary collaboration, and developing appropriate policy and governance frameworks. Large-scale demonstrations and field trials are crucial for validating the effectiveness, feasibility, and scalability of integrated CDR technology and AI systems.

By addressing these aspects and focusing on future research directions, the integration of AI and CDR technology can significantly contribute to mitigating climate change, reducing greenhouse gas emissions, and building a sustainable future. Advanced technology and innovative approaches will play a crucial role in achieving carbon reduction targets and creating a more environmentally friendly world.

**Author Contributions:** Writing—original draft preparation, G.L. and T.L.; methodology, Z.L. and S.W.; software, R.L. and C.S.; validation, C.Z. All authors have read and agreed to the published version of the manuscript.

**Funding:** Thanks to the support of the Science and Technology Project of the State Grid Corporation of China: 4000-202316071A-1-1-ZN ≪Research on New Energy Storage Dispatch Operation Mechanism and Method Considering New Energy Consumption and Power Grid Support≫.

**Data Availability Statement:** Data available on request from the authors.

**Conflicts of Interest:** The authors declare that there are no conflict of interest.

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
