# Peer review of "Integration of Carbon Dioxide Removal (CDR) Technology and Artificial Intelligence (AI) in Energy System Optimization"

_processes, doi:10.3390/pr12020402_

Round 1

Reviewer 1 Report

Comments and Suggestions for Authors

the authors are introducing a review of research in the field of climate change and carbon dioxide emissions reductions

the authors are focusing on the integration between AI and CDR for energy system optimization

in genral the paper is well written and clear, however the following should be considerd:

1. the Abstract is not clear, it is somehow distorting. not showing clearly the sauthors target

2. the authors keep using sentences with the same meaning without any need (please revise section 1.3)

3. authors are speaking about a gap in the reviews as it did not focus on the integration of Ai and CDR, is that the real contribution of the review? 

4. the summarized 4 approaches for integrating CDR and AI are good, however i believe some more approaches exist. why did the authors focused on these approaches only

5. the research directions suggested in the review are already known, some publications are already there in these directions. would you explain the paper contribution in that aspect

6. conclusion is too long and again repeating what was given in the paper itself, it needs to be rephrased

7. abbreviations list must be included in tthe very beginning

Reviewer 2 Report

Comments and Suggestions for Authors

There are major concerns that must be reconsidered to improve the content of a paper comprehensively. Nevertheless, the reviewer hopes the authors will find the below-given comments helpful to enhance their study:

·       The reviewer thinks that the authors give excessively detailed information to publish as a scientific review paper. This can be acceptable for books chapter or other publications but it’s not suitable for journal publication. The Authors should recheck the study and shorten the paper.

·       Sections that are repeated several times within the different parts of the study and have the same meaning should be removed and simplified. For example, the benefits of AI have been repeatedly written with similar meanings.

·       The authors said, "Existing reviews have often focused on either CDR technologies or AI applications in energy systems without providing a comprehensive analysis of their combined potential” but several studies found about this subject in the literature are examined both of them together. Authors should recheck their literature searches.

·       Reviewer found some of the expressions like “We discuss their impact ….” or “We explore their potential ……” written in non-scientific language. The authors should check the study and correct the writing style.

·       Authors should rearrange Figure 1 with a clear explanation. Long explanation sentences written on the figure make it difficult to understand. Instead, explanations should be given in paragraphs with reference to the figure.

·       Figures 3, 4, 6, and 8 consist of visuals that are too simple to be included in a scientific article. These shapes need to be recreated by the Authors.

·       Authors should summarize the technologies they explain in Section 1.2 with a table showing their advantages and disadvantages comparatively.

Comments on the Quality of English Language

The English of the paper is good, but a native speaker should recheck it. And some of the writing errors such as “approoches baesd” in line 25, “intergrationin line 30 should be corrected.

Reviewer 3 Report

Comments and Suggestions for Authors

The manuscript discusses the integration of carbon dioxide removal technology into an optimized energy system. The structure is well designed and looks good to me. I have a few comments and suggestions for authors:

1- If possible based on your study provide the quantitative measure of applying CDR with AI in energy systems (the improvement of CDR with AI).
2-Please provide bullet points in the gaps in knowledge to make it easier to follow (sections 1-3). Why do we need your review, and what aspects will be covered by your study? 
3-Please check the title of 3.3, and correct it if necessary.  

I hope these comments will be helpful to authors and Processes Journal readers.

Comments on the Quality of English Language

The English language is fine. 

Round 2

Reviewer 1 Report

Comments and Suggestions for Authors

Good job, I have no more comments 

Reviewer 2 Report

Comments and Suggestions for Authors

The Reviewer thanks the authors for their detailed corrections. The study is suitable for publication in this revised version.